# Systematic Generalization with Edge Transformers

**Leon Bergen**
University of California, San Diego
`lbergen@ucsd.edu`

**Timothy J. O'Donnell**
McGill University
Quebec Artificial Intelligence Institute (Mila)
Canada CIFAR AI Chair

**Dzmitry Bahdanau**
Element AI, a ServiceNow company
McGill University
Quebec Artificial Intelligence Institute (Mila)
Canada CIFAR AI Chair

## Abstract

Recent research suggests that systematic generalization in natural language understanding remains a challenge for state-of-the-art neural models such as Transformers and Graph Neural Networks. To tackle this challenge, we propose Edge Transformer, a new model that combines inspiration from Transformers and rule-based symbolic AI. The first key idea in Edge Transformers is to associate vector states with every edge, that is, with every pair of input nodes—as opposed to just every node, as it is done in the Transformer model. The second major innovation is a *triangular attention* mechanism that updates edge representations in a way that is inspired by unification from logic programming. We evaluate Edge Transformer on compositional generalization benchmarks in relational reasoning, semantic parsing, and dependency parsing[1]. In all three settings, the Edge Transformer outperforms Relation-aware, Universal and classical Transformer baselines.

## 1 Introduction

Transformers (Vaswani et al., 2017) have become ubiquitous in natural language processing and deep learning more generally (e.g., Devlin et al., 2018; Raffel et al., 2020; Carion et al., 2020). Nevertheless, systematic (or compositional) generalization benchmarks remain challenging for this class of models, including large instances with extensive pre-training (Keysers et al., 2020; Tsarkov et al., 2020; Gontier et al., 2020; Furrer et al., 2020). Similarly, despite their increasing application to a variety of reasoning and inference problems, the ability of Graph Neural Networks' (GNN) (Gori et al., 2005; Scarselli et al., 2009; Veličković et al., 2018) to generalize systematically has also been recently called into question (Sinha et al., 2019, 2020). Addressing these challenges to systematic generalization is critical both for robust language understanding and for reasoning from other kinds of knowledge bases.

In this work, inspired by symbolic AI, we seek to equip Transformers with additional representational and computational capacity to better capture the kinds of information processing that underlie systematicity. Our intuition is that classical Transformers can be interpreted as a kind of inferential system similar to a continuous relaxation of a subset of the Prolog logic programming language. In particular, as we discuss in the next section, Transformers can be seen as implementing a kind of reasoning over *properties* of entities—for example, RED($\cdot$) or TABLE($\cdot$)—where any *relations*

---

[1]Code for our experiments can be found at: github.com/bergen/EdgeTransformer

35th Conference on Neural Information Processing Systems (NeurIPS 2021).

required for such reasoning—for example PART-OF$(\cdot, \cdot)$ or GRANDMOTHER$(\cdot, \cdot)$—are inferred on the fly with the self-attention mechanism.

Building on this intuition, in this work we propose *Edge Transformers*, a generalization of the Transformer model that uses a novel *triangular attention* mechanism that is inspired by a much more general family of inferential rules.

To achieve this, we endow the Edge Transformer with a 3D tensor state such that every *edge*, that is, every pair of input nodes, contains a vector that represents relations between the two nodes. The updates of each edge are computed using all adjacent edges in a way that is directly inspired by unification in logic programming. While the use of edge features or even dynamic edge states can be found in the Transformer and GNN literatures (Shaw et al., 2018; Gilmer et al., 2017; Gong and Cheng, 2019), to the best of our knowledge such triangular updates are novel to our approach.

We evaluate the Edge Transformers on three recently proposed compositional generalization challenges. First, on the graph version of the CLUTRR relational reasoning challenge (Sinha et al., 2019), our model displays stronger systematic generalization than Graph Attention Networks (Veličković et al., 2018) and Relation-aware Transformers (Shaw et al., 2018). Second, on both dependency parsing and semantic parsing versions (Goodwin et al., 2021a) of the Compositional Freebase Questions benchmark (Keysers et al., 2020) our model achieves a higher parsing accuracy than that of classical Transformers and BiLSTMs. Last but not least, Edge Transformer achieves state-of-the-start performance of 87.4% on the COGS benchmark for compositional generalization in semantic parsing.

## 2  Intuitions

A Transformer operates over a set of $n$ entities—such as words in a sentence—which we represent as *nodes* in a graph like those displayed on the left-hand side of Figure 1. Each node is associated with a $d$-dimensional *node-state* $X(i)$ for $i \in n$, which can be thought of as the output associated with applying some function to the node—that, a representation of node *properties*. In transformers, computation proceeds by sequentially associating each node with a number $l$ of node-states, with each state updated from node states at the preceding layer via the attention mechanism.

Adopting a Prolog-like notation, we could write the fundamental inferential process implemented by the transformer architecture as

$$X^{l+1}(1) \vdash_{\mathcal{A}} X^l(1), X^l(2), \ldots, X^l(n). \tag{1}$$

In logic programming, the turnstile symbol $\vdash$ means that whenever the right-hand side of the expression is true, the left-hand side must be true. In our interpretation of Transformers, the inference rules expressed by $\vdash_{\mathcal{A}}$ are learned via the attention mechanism $\mathcal{A}$, as is the meaning of each property $X^l(\cdot)$. Classical transformers can therefore be interpreted as an architecture specialized to learning how entity properties can be inferred from the properties of other entities.

Despite the power of this mechanism, it still has noteworthy limitations. In particular, for many practical reasoning and NLU tasks systems must learn dependencies between *relations*, such as family relations: MOTHER$(x, y)$ and MOTHER$(y, z)$ implies GRANDMOTHER$(x, z)$. Such a general reasoning problem might be expressed in Prolog-like notation as follows.

$$X^{l+1}(1, 2) \vdash_{\mathcal{A}} X^l(1, 1), X^l(1, 2), \ldots, X^l(2, 1), X^l(2, 2), \ldots. \tag{2}$$

It is of course possible for classical transformers to capture such reasoning patterns. But for them to do so, each transformer state $X^{l+1}(1)$ must encode **both** the properties of the relation itself, that is, MOTHER$(\cdot, \cdot)$ versus GRANDMOTHER$(\cdot, \cdot)$—as well as all target nodes $x$ with which node 1 stands in the relation, that is MOTHER$(1, x)$. In other words, to encode relations a classical transformer must use its state representations for two distinct purposes. This mixing of concerns places a high burden on the learning, and we hypothesize that the resulting inductive bias hinders the ability of this class of models to generalize systematically.

To address this problem we propose that the fundamental object represented by the model should not be a state associated with a node but, rather, states associated with edges, like those displayed on the right of Figure 1. Such edge representations in turn are updated based on an attention mechanism which attends to other edges. This mechanism is inspired by the process of unification in logic programming. Critical to an inference rule such as

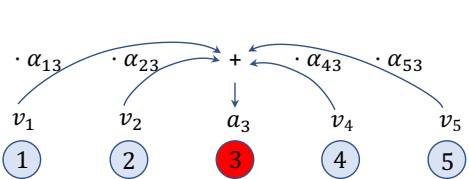 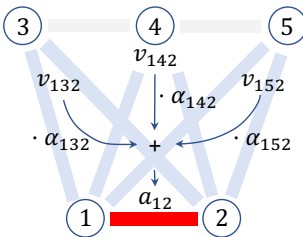

Figure 1: Left: Transformer self-attention computes an update $a_3$ for the node 3 by mixing value vectors $v_i$ associated with each node $i$ with attention weights $\alpha_i$. Right: triangular attention computes an update $a_{12}$ for the edge $(1, 2)$ by mixing value vectors $v_{1l2}$ computed for each triangle $(1, l, 2)$ with attention weights $\alpha_{1l2}$. Values $v_3, v_{112}, v_{122}$ and their contributions are not shown for simplicity.

GRANDMOTHER$(x, y) \vdash_{\mathcal{A}}$ MOTHER$(x, z)$ MOTHER$(z, y)$ is the fact that the two predicates on the right hand side of this rule share a variable $z$. Successfully inferring GRANDMOTHER$(x, y)$ which involve finding a binding for $z$ which satisfies both MOTHER$(z, y)$ and MOTHER$(x, z)$—a process which is handled by *unification* in logic programming. We believe that this is a fundamental aspect of relational reasoning and to capture this we make use of a form of attention which we call *triangular attention* and which we further describe in the next section.

## 3 Edge Transformers

An edge transformer operates on a complete graph with $n$ nodes and $n^2$ directed edges (we allow self-connections) labeled with $d$-dimensional feature vectors. An edge transformer state therefore is a 3-dimensional tensor $X \in \mathbb{R}^{n,n,d}$ that consist of edges states $X(i, j)$ corresponding to every edge $(i, j)$, $i, j \in [1, n]$. We will also write $x_{ij}$ for each edge state. An edge transformer layer computes

$$X' = \text{FFN}(\text{LN}(X + \text{TriangularAttention}(\text{LN}(X)))). \tag{3}$$

Here, FFN$(X)$ is a fully-connected feed-forward network with one hidden layer of $4 \cdot d$ units that is applied independently to each edge state $x_{ij}$, LN stands for the layer normalization mechanism (Ba et al., 2016) which rescales activations of each feature $f \in [1, d]$ across its occurrences in edge states $x_{ij}$. Equation 3 follows the familiar structure of transformer (Vaswani et al., 2017) computations with two important differences: (a) It uses a 3D-tensor state instead of a matrix state; and (b) it makes use of a novel triangular attention mechanism that we describe below.

For a single edge state $x_{ij}$ a single-head triangular attention update outputs a vector $a_{ij}$ that is computed as follows:

$$a_{ij} = W^o \sum_{l \in [1,n]} \alpha_{ilj} v_{ilj}, \tag{4}$$

$$\alpha_{ilj} = \underset{l \in [1,n]}{\text{softmax}} \, q_{il} k_{lj} / \sqrt{d}, \tag{5}$$

$$q_{il} = W^q x_{il}, \tag{6}$$

$$k_{lj} = W^k x_{lj}, \tag{7}$$

$$v_{ilj} = V^1 x_{il} \odot V^2 x_{lj}, \tag{8}$$

where $\odot$ stands for elementwise multiplication, and $W^q, W^k, W^o, V^1, V^2 \in \mathbb{R}^{d,d}$ are matrices that are used to produce key, query, output and value vectors $k_{il}, q_{lj}, a_{ij}$ and $v_{ilj}$ respectively. Here and in the rest of the paper we omit bias terms for clarity.

Informally, updates associated with an edge $(i, j)$ proceed by aggregating information across all pairs of edges that share a node $l$, that is, all pairs $(i, l)$ and $(l, j)$. The updated edge value, $a_{ij}$, is an attention-weighted mixture of contributions from each such pair of edges. Figure 1 visualizes some of the key differences between transformer and edge transformer computations. Note that edges $(i, j)$, $(i, l)$, and $(l, j)$ form a triangle in the figure—hence the name of our attention mechanism.

We define a multi-head generalization of the above mechanism in a way similar to the implementation of multi-head attention in vanilla transformers. Specifically, each head $h \in [1, m]$ will perform Edge

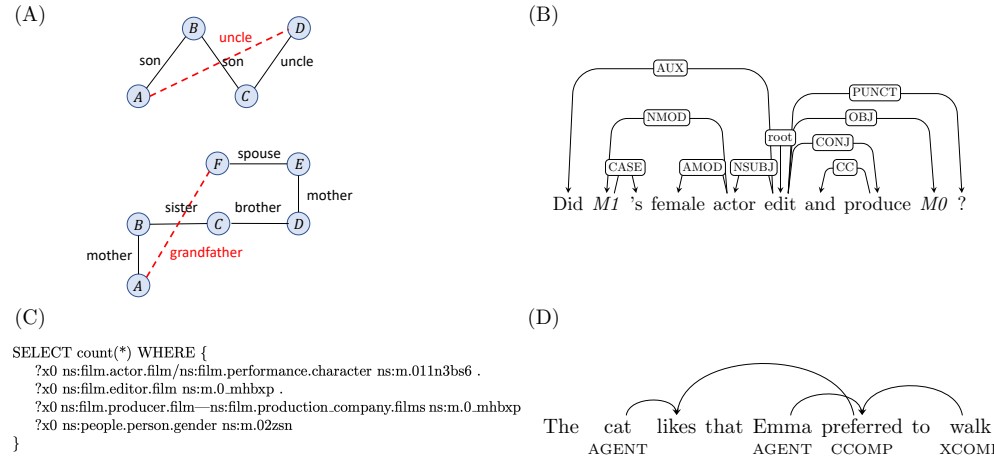

Figure 2: Illustrations of the benchmarks that we use to evaluate the Edge Transformer. **(A)** Relational reasoning scenarios for the CLUTRR benchmark, with the relation length of 3 (top) and 6 (bottom). **(B)** Dependency parsing for Compositional Freebase Questions (CFQ; reproduced with permission, Goodwin et al., 2021b). **(C)** SPARQL query and the corresponding question from the CFQ semantic parsing dataset. **(D)** graph representations of COGS semantic parses (Ontanón et al., 2021).

Attention using the smaller corresponding matrices $W^q, W^k, V^1, V^2 \in \mathbb{R}^{d,d/m}$, and the joint output $a_{ij}$ will be computed by multiplying a concatenation of the heads' output by $W^o$:

$$a_{ij} = W^o \left[ a_{ij}^1; \ldots; a_{ij}^m \right]. \tag{9}$$

In practice a convenient way to implement an efficient batched version of edge attention is using the Einstein summation operation which is readily available in modern deep learning frameworks.

**Tied & Untied Edge Transformer**   Edge Transformer layers can be stacked with or without tying weights across layers. By default we tie the weights in a way similar to Universal Transformer (Dehghani et al., 2019). We will refer to the version with weights untied as Untied Edge Transformer.

**Input & Output**   The Edge Transformer's initial state $X^0$ is defined differently for different applications. When the input is a labeled graph with (possibly null) edge labels, the initial state $x_{ij}^0$ is produced by embedding the labels with a trainable embedding layer. When the input is a sequence, such as for example a sequence of words or tokens $t_i$, self-edges $x_{ii}^0$ are initialized with the embedding for token $t_i$, and all edges $x_{ij}^0$ are initialized with a relative position embedding (Shaw et al., 2018):

$$x_{ij}^0 = \begin{cases} e(t_i) + a(0) & \text{if } i = j \\ a(i-j) & \text{if } i \neq j \end{cases} \tag{10}$$

where $e$ is a trainable $d$-dimensional embedding layer, and $a(i-j)$ is a relative position embedding for the position difference $i - j$.

To produce a graph-structured output with Edge Transformer, a linear layer with cross-entropy loss can be used for every edge $(i, j)$. For classification problems with a single label for the entire input the label can be associated with one of the edges, or edge representations can be pooled.

Lastly, an encoder-decoder version of Edge Transformer can be used to perform sequence-to-sequence transduction. Our approach follows that used with the original Transformer architecture (Vaswani et al., 2017). First, an encoder Edge Transformer processes the question and produces a 3D tensor

Table 1: Hyperparameter settings for the Edge Transformer and for the baselines. $L$ is the number of layers, $d$ is the dimensionality, $h$ is the number of heads, $B$ is the batch size, $\rho$ is the learning rate, $T$ is training duration. For CFQ, "dep." stands for dependency parsing and "sem." stands for semantic parsing.

| Model | Task | $L$ | $d$ | $h$ | $B$ | $\rho$ | $T$ |
|---|---|---|---|---|---|---|---|
| Edge Transformer | CLUTRR | 8 | 200 | 4 | 400 | $1 \cdot 10^{-3}$ | 50 epochs |
| RAT | CLUTRR | 8 | 320 | 8 | 200 | $1 \cdot 10^{-3}$ | 50 epochs |
| RRAT | CLUTRR | 6 | 320 | 8 | 200 | $1 \cdot 10^{-3}$ | 50 epochs |
| Edge Transformer | CFQ dep. | 7 | 360 | 4 | $5 \cdot 10^2$ words | $1 \cdot 10^{-3}$ | 8000 steps |
| Transformer | CFQ dep. | 7 | 360 | 4 | $1 \cdot 10^3$ words | $1 \cdot 10^{-3}$ | 8000 steps |
| Universal Transformer | CFQ dep. | 8 | 360 | 4 | $5 \cdot 10^2$ words | $1 \cdot 10^{-3}$ | 8000 steps |
| Edge Transformer | CFQ sem. | 6 | 256 | 8 | 64 | $6 \cdot 10^{-4}$ | 100 epochs |
| Universal Transformer | CFQ sem. | 4 | 256 | 8 | 64 | $6 \cdot 10^{-4}$ | 100 epochs |
| Edge Transformer | COGS | 3 | 64 | 4 | 100 | $5 \cdot 10^{-4}$ | 200 epochs |

state $x_{enc}$. Next, the decoder Edge Transformer generates output tokens one by one, left-to-right. To produce the output distribution for the token we feed the state of the loop edge $(i, i)$ through a linear layer. To connect the decoder to the encoder we insert $x_{enc}$ into the decoder's initial state. This operation is the Edge Transformer's equivalent to how the Transformer decoder cross-attends to the Transformer encoder's hidden states. To compute training log-likelihoods we adapt the Transformer decoder's causal masking approach to our triangular attention mechanism.

## 4 Experiments

In our experiments we compare the systematic generalization ability of Edge Transformers to that of Transformers (Vaswani et al., 2017), Universal Transformers (Dehghani et al., 2019), Relation-aware Transformers (Shaw et al., 2018), Graph Attention Networks (Veličković et al., 2018) and other baselines. We focus on three synthetic benchmarks with carefully controlled train-test splits, Compositional Language Understanding and Text-based Relational Reasoning (CLUTRR), proposed by Sinha et al. (2019), Compositional Freebase Questions (CFQ) proposed by Keysers et al. (2020) and Compositional Generation Challenge based on Semantic Interpretation (COGS) by Kim and Linzen (2020). Systematic generalization refers to the ability to recombine known primitive units in novel combinations. These three benchmarks match test and train in terms of primitive atoms used and rules of combination, but test on novel combinations of atoms not seen in train. These benchmarks are thus appropriate for assessing whether Edge Transformers have a greater ability to generalize systematically than existing models.

In all tables we report mean $\pm$ standard error over multiple runs. For details on the hyperparameter search procedure see Appendix A. For the chosen hyperparameter settings see Table 1.

### 4.1 Relational Reasoning on Graphs

In the first round of experiments, we use the CLUTRR benchmark proposed by Sinha et al. (2019). CLUTRR evaluates the ability of models to infer unknown familial relations between individuals based on sets of given relations. For example, if one knows that A is a son of B, C is a son of B and C has an uncle called D, one can infer that D must also be the uncle of A (see Figure 2, A). The authors propose training models on scenarios that require a small number (2, 3 or 4) of supporting relations to infer the target relation and test on scenarios where more (up to 10) supporting relations are required. We will refer to the number of inferential steps required to prove the target relation as the *relation length* $k$. For relation lengths $k = 2, 3, 4$, we train models on the original CLUTRR training set from Sinha et al. (2019), which contains 15k examples. For relation lengths $k = 2, 3$, we generate a larger training set containing $35k$ examples using Sinha et al. (2019)'s original code, allowing us to measure systematic generalization performance with less variance. We use the noiseless (only the required facts are included) graph-based version of the challenge, where the input is represented as a labeled graph, and the model's task is to label a missing edge in this graph.

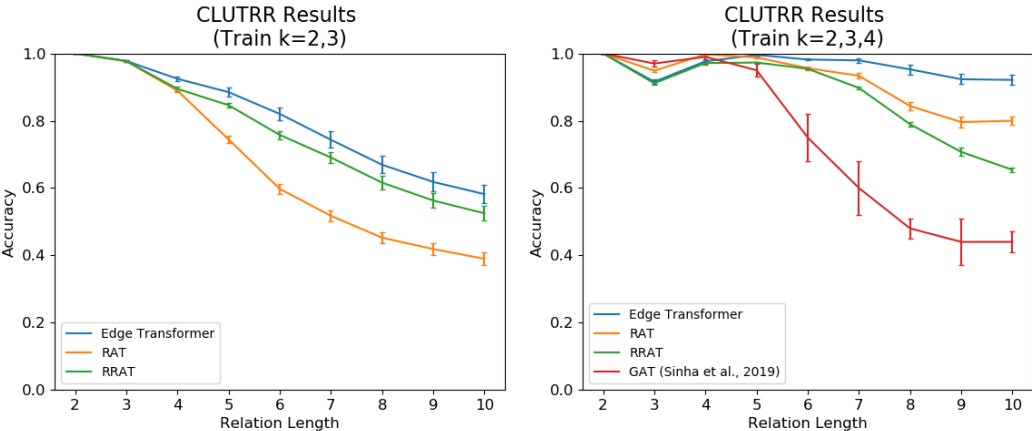

Figure 3: Relation prediction accuracy on CLUTRR for test relations of lengths $k \in [2, 10]$. GAT results from Sinha et al. (2019) (only shown for the $k = 2, 3, 4$ task, as the $k = 2, 3$ training set was regenerated). Edge, Relation-Aware (RAT), Relation-updating Relation-aware (RRAT) Transformer results are averaged over 10 runs.

We create the Edge Transformer's initial state $X^0$ by embedding edge labels. We apply a linear layer to the final representation $x_{ij}^L$ of the query edge $(i, j)$ in order to produce logits for a cross-entropy loss. Our main baseline is a Relation-aware Transformer (RAT) (RT; Shaw et al., 2018), a Transformer variant that can be conditioned on arbitrary graphs. Edges for RAT are initialized by embedding edge labels, and nodes are initialized with zero vectors. Logits are computed by applying a linear layer to concatenated representations of the nodes that occur in the queried edge. We also compare against a variant of RAT which updates both nodes and edges. For this Relation-updating Relation-aware Transformer (RRAT), node updates are computed the same way as in RAT. The update for the edge $x_{ij}$ is computed by concatenating the representations for nodes $i$ and $j$, and applying a linear layer. Separate feed-forward layers are applied to nodes and edges. Logits are computed by applying a linear layer to the queried edge. Finally, for the $k = 2, 3, 4$ setting we include the Graph Attention Networks (GAT, Veličković et al. (2018)) results from the original CLUTRR paper for comparison. Figure 3 displays the results. One can see the Edge Transformer beats all baselines by wide margins, in both $k = 2, 3$ and $k = 2, 3, 4$ training settings.

### 4.2 Dependency Parsing of CFQ

In our second set of experiments we use the dependency parsing version of the CFQ challenge (Keysers et al., 2020) recently proposed by Goodwin et al. (2021b). The original CFQ challenge evaluates compositional generalization in semantic parsing—in the case of CFQ, the task of translating natural language questions into corresponding knowledge graph queries. The CFQ benchmark was constructed by generating test/train splits which minimize the difference between the test and train distributions over primitive units, like words, while maximizing the *compound divergence*—the dissimilarity between test and train distributions over larger structures, like phrases. As a result, at test time, a model has to generalize to novel syntactic constructions, such as "Was NP NP?" ("Was Alice the sister of Bob?") that were not seen during training. Keysers et al. (2020) released three different maximum compound divergence (MCD) splits. Goodwin et al. (2021b) propose a dependency parsing version of the CFQ task and show that the MCD splits are challenging for a state-of-the-art dependency parser (see Figure 2, B). This dependency parsing task is convenient for evaluating Edge Transformer in an encoder-only graph prediction setting that is simpler than the original sequence-to-sequence formulation of the CFQ task (see Section 4.3 for results in the original setting).

We use the Stanza framework for dependency parsing (Qi et al., 2020) in which we replace the default BiLSTM model with Transformer, Universal Transformer or Edge Transformer variants. The models are trained to label each pair of positions $i$ and $j$ with a dependency type label or a null label. These predictions are then combined into a parse tree by Stanza's spanning-tree-based parsing algorithm. The model's initial state $X^0$ is initialized from token and relational embeddings as

Table 2: Dependency parsing accuracy on Compositional Freebase Queries (CFQ). EM is Exact Match accuracy, LAS is Labeled Attachment Score. BiLSTM results are from Goodwin et al. (2021a). Transformer, Universal Transformer and Edge Transformer results are averaged over 10 runs.

| Model | MCD1 | | MCD2 | | MCD3 | |
|---|---|---|---|---|---|---|
| | EM | LAS | EM | LAS | EM | LAS |
| BiLSTM | $96.6 \pm 1.3$ | $\mathbf{99.6} \pm 0.2$ | $71.4 \pm 2.6$ | $93.3 \pm 0.9$ | $56.8 \pm 2.8$ | $90.9 \pm 1.0$ |
| Transformer | $75.3 \pm 1.7$ | $97.0 \pm 0.1$ | $59.3 \pm 2.7$ | $91.8 \pm 0.4$ | $48.0 \pm 1.6$ | $89.4 \pm 0.3$ |
| Universal Transformer | $80.1 \pm 1.7$ | $97.8 \pm 0.2$ | $68.6 \pm 2.3$ | $92.5 \pm 0.4$ | $59.4 \pm 2.0$ | $90.5 \pm 0.5$ |
| Edge Transformer | $\mathbf{97.4} \pm 0.8$ | $99.4 \pm 0.2$ | $\mathbf{89.5} \pm 0.5$ | $\mathbf{96.8} \pm 0.1$ | $\mathbf{89.1} \pm 0.2$ | $\mathbf{96.3} \pm 0.1$ |

described by Equation 10. For Edge Transformer, pairwise output labels between positions $i$ and $j$ are predicted directly from edge $x_{ij}$. For the baselines, these labels are obtained by passing each pair of nodes through a biaffine layer, following Dozat and Manning (2017). The training set contains approximately 100K questions.

We report the Labeled Attachment Score (LAS) that assesses partial correctness (F1) of the predicted dependency trees as well as the exact match accuracy (EM) that only takes into account completely predicted trees. As can be seen in Table 2, Edge Transformer enjoys a sizable performance advantage (17-30% EM accuracy) over all baseline models on the MCD2 and MCD3 splits. On MCD1 split both the BiLSTM and Edge Transformer achieve near-perfect performance, while the classical Transformer lags behind.

## 4.3 Semantic Parsing of CFQ

We furthermore experiment with the original semantic parsing task from CFQ. In this task, natural language questions must be translated to knowledge graph queries in SPARQL query language (see Figure 2, C). We use the encoder-decoder version of Edge Transformer to produce the sequence of query tokens from a sequence of questions tokens.

The $O(n^3)$ computational complexity and memory footprint of Edge Transformer pose a challenge in this round of experiments as the decoder's input size $n$ is larger than for the other tasks (more than 100 tokens for some examples). To remedy this, we filter out training examples in which the SPARQL queries are more than 50 tokens long. Even after this filtering training an Edge Transformer model on CFQ semantic parsing requires 1-2 days of using 4 NVIDIA V100 GPUs. This makes a larger scale hyperparameter search prohibitively expensive for this model. To tune Edge Transformer, we first perform the hyperparameter search for the computationally cheaper Universal Transformer baseline which we found to outperform the vanilla Transformer in preliminary experiments. For Edge Transformer we only try a few variations on the chosen Universal Transformer configuration (see Appendix A.2 for more details). As suggested in the original CFQ paper (Keysers et al., 2020), we tune hyperparameters on the random split and keep the compositional generalization splits for testing. We reduce the size of the random split training data from $\sim 90K$ examples to $\sim 20K$ to increase the sensitivity to different hyperparameter choices. For each MCD split we report mean exact match accuracy for 30 Universal Transformer and 5 Edge Transformer runs.

The results are shown in Table 3. Edge Transformer reaches the state-of-the-art performance among the non-pretrained encoder-decoder models. It reaches $24.67 \pm 1.27$ % average MCD accuracy that approximately 3 percentage points ahead of comparable competition. Notably, Edge Transformer also beats Universal Transformer on our filtered random split ($99.08 \pm 0.1$% vs $98.45 \pm 0.08$ % accuracy).

## 4.4 Semantic Parsing of COGS

As our final evaluation of the ability of Edge Transformers to generalize systematically, we make use of the COGS benchmark of Kim and Linzen (2020). COGS defines a semantic parsing task from natural language sentences to linguistic denotations expressed in a logical formalism derived from that of Reddy et al. (2017). The training/test split in COGS is designed to test the ability of models to generalize from observed sentences, to sentences of greater complexity that exhibit novel combinations of primitive words and phrases. We train and test on the graph representation of the

Table 3: Exact match accuracy of the produced SPARQL queries for CFQ semantic parsing experiments. ♠ and ◇ denote results from (Keysers et al., 2020) and (Furrer et al., 2020) respectively. We convert the 95% confidence intervals reported in both papers to standard errors.

| Model | MCD avg. | MCD1 | MCD2 | MCD3 |
|---|---|---|---|---|
| Universal Transformer | $21.26 \pm 1.06$ | $42.7 \pm 2.55$ | $9.46 \pm 1.2$ | $11.62 \pm 0.68$ |
| **Edge Transformer** | $\mathbf{24.69} \pm \mathbf{1.27}$ | $\mathbf{47.73} \pm \mathbf{3.63}$ | $\mathbf{13.14} \pm \mathbf{2.12}$ | $\mathbf{13.2} \pm \mathbf{1.38}$ |
| Universal Transformer ♠ | $18.9 \pm 1.1$ | $37.4 \pm 1.8$ | $8.1 \pm 1.3$ | $11.3 \pm 0.2$ |
| Evolved Transformer ◇ | $20.8 \pm 0.6$ | $42.4 \pm 0.8$ | $9.3 \pm 0.6$ | $10.8 \pm 0.2$ |
| T5-small ◇ | $21.4 \pm 1.2$ | $42.5 \pm 2.1$ | $11.2 \pm 1.2$ | $10.6 \pm 0.3$ |

Table 4: Top section: graph prediction accuracy for Edge Transformer and comparable graph prediction Transformer results by Ontanón et al. (2021) (marked by ◇). Bottom section: best COGS results by Akyürek et al. (2021) (♠) and Csordás et al. (2021) (♣) obtained by predicting semantic parses as sequences of tokens.

| Model | Generalization Accuracy |
|---|---|
| **Edge Transformer** | $\mathbf{87.4} \pm 0.4$ |
| Universal Transformer - Attention Decoder ◇ | 78.4 |
| LSTM+Lex ♠ | $82 \pm 1.$ |
| Transformer ♣ | $81 \pm 0.01$ |

COGS semantic parses provided by Ontanón et al. (2021). The nodes in these semantic graphs are input tokens that are labeled with word-level category information, for example, semantic role and part-of-speech. The edges connect some pairs of nodes to indicate parent-child dependencies between them, similar to a dependency parse (see Figure 2,D). The Edge Transformer model predicts node labels for token $i$ by applying several linear layers (one for each label type) to self-edges $x_{ii}$. For edge labels, each node has a unique parent (for nodes with no parent, we follow Ontanón et al. (2021) and treat them as being their own parent). To predict the parent of node $i$, we apply a linear layer to all edges $x_{ji}$, and take a softmax across the position axis. Hyperparameters for Edge Transformer were not tuned for this task. Instead, we used the best setting identified by Ontanón et al. (2021) for the Transformer architecture (see Appendix A.3), and default settings for the optimizer.

As can be seen from Table 4, Edge Transformers exhibits a state-of-the-art performance of $87.4 \pm 0.4$ % on COGS, outperforming the baseline graph prediction models from (Ontanón et al., 2021) by 9 percentage points on this challenging task. This performance is also higher than what is reported in the literature for the original sequence prediction version of the COGS task, including 82% accuracy of the lexicon-equipped LSTM (LSTM+Lex by Akyürek et al. (2021)) and 81% accuracy of the carefully initialized Transformer model (Csordás et al., 2021).

## 4.5 Ablation Experiments

To explore the performance of Edge Transformers, we experiment with several variations of the model. First, we train Edge Transformers with untied weights, using separate weights for each layer, as is usually done with classical Transformers. We refer to this variant as *untied weights*. Second, we simplify the Edge Transformer computations by computing the value $v_{ilj}$ for each triangle $(i, l, j)$ using only the edge $(i, l)$ instead of using both edges $(i, l)$ and $(l, j)$. This is equivalent to replacing Equation 8 with $v_{ilj} = V^1 x_{il}$. We will refer to this condition as *value ablation*. Third and finally, we use a key vector $k_{ij} = W^k x_{ij}$ instead of $k_{lj} = W^k x_{lj}$ for computing attention scores in Equation 5. We call this condition *attention ablation*.

We compare our base model to the above variants on the CLUTRR task, the MCD3 split of the CFQ dependency parsing task and on the COGS task. In CLUTTR experiments we rerun the hyperparameters for each ablated model, whereas in other experiments we keep the same hyperparameters as for base Edge Transformer. Table 5 shows results for CLUTRR and Table 6 shows results for CFQ and COGS. One can see that value ablation consistently leads to worse performance on CLUTRR and CFQ dependency parsing, confirming the importance of the unification-inspired triangular updates described in Equations 5-8 and illustrated in Figure 1. Likewise, attention ablation hurts the perfor-

Table 5: Results of ablation experiments on CLUTRR. "Base" is Edge Transformer as described in Section 3. See Section 4 for details on ablated model variants. Results are averaged over 10 runs.

| Model | Test k | | | | |
| | 6 | 7 | 8 | 9 | 10 |
|---|---|---|---|---|---|
| **Train on k=2,3** | | | | | |
| Base | $82.0 \pm 2.0$ | $74.3 \pm 2.4$ | $66.9 \pm 2.6$ | $61.8 \pm 2.9$ | $58.1 \pm 2.8$ |
| Untied weights | $78.9 \pm 1.0$ | $71.0 \pm 1.1$ | $63.2 \pm 1.1$ | $58.5 \pm 1.2$ | $55.3 \pm 1.2$ |
| Value ablation | $54.4 \pm 3.4$ | $44.2 \pm 3.6$ | $36.9 \pm 3.6$ | $33.9 \pm 3.7$ | $30.7 \pm 3.6$ |
| Attention ablation | $84.4 \pm 1.7$ | $73.1 \pm 2.3$ | $63.2 \pm 2.2$ | $56.4 \pm 2.1$ | $51.4 \pm 1.9$ |
| **Train on k=2,3,4** | | | | | |
| Base | $98.1 \pm 0.3$ | $97.9 \pm 0.7$ | $95.3 \pm 1.5$ | $92.3 \pm 1.5$ | $92.1 \pm 1.5$ |
| Untied weights | $98.2 \pm 0.3$ | $98.2 \pm 0.3$ | $95.6 \pm 0.7$ | $93.1 \pm 1.4$ | $90.2 \pm 1.4$ |
| Value ablation | $92.6 \pm 1.3$ | $86.2 \pm 2.1$ | $82.5 \pm 3.2$ | $77.1 \pm 3.7$ | $75.2 \pm 2.7$ |
| Attention ablation | $95.9 \pm 2.1$ | $95.2 \pm 2.5$ | $92.8 \pm 3.1$ | $89.8 \pm 2.9$ | $78.7 \pm 2.5$ |

Table 6: Results of ablation experiments on CFQ dependency parsing and COGS graph semantic parsing. "Base" stands for the Edge Transformer as described in Section 3. See Section 4 for details on ablated model variants. Results are averaged over 10 runs. EM is Exact Match accuracy, LAS is Labeled Attachment Score.

| Model | MCD3 | | COGS |
| | EM | LAS | Gen. Accuracy |
|---|---|---|---|
| Base | $89.1 \pm 0.2$ | $96.3 \pm 0.1$ | $87.4 \pm 0.4$ |
| Untied weights | $37.2 \pm 12.4$ | $53.2 \pm 14.2$ | $86.3 \pm 0.8$ |
| Value ablation | $84.5 \pm 1.1$ | $95.5 \pm 0.3$ | $87.8 \pm 0.5$ |
| Attention ablation | $88.1 \pm 0.7$ | $96.4 \pm 0.1$ | $86.3 \pm 0.7$ |

mance on CLUTRR and COGS. Lastly, for untying weights, there is some evidence of deterioration for the CLUTRR setting where we train on $k = 2, 3$, and a large effect for CFQ dependency parsing.

## 5  Related Work

Most variations of the Transformer architecture by Vaswani et al. (2017) aim to improve upon its quadratic memory and computation requirements. To this end, researchers have experimented with recurrency (Dai et al., 2019), sparse connectivity (Ainslie et al., 2020), and kernel-based approximations to attention (Choromanski et al., 2021). Most relevant here is the use of relative positional embeddings in Relation-aware Transformers proposed by (Shaw et al., 2018) and refined by (Dai et al., 2019). While this approach introduces edges to Transformers, unlike in Edge Transformers, these edges are static; that is, they do not depend on external input other than the respective edge labels.

Graph Neural Networks (GNNs; Gori et al., 2005; Scarselli et al., 2009; Kipf and Welling, 2017), and especially Graph Attention Networks (Veličković et al., 2018) are closely related to Transformers, especially relation-aware variants. Unlike Transformers, GNNs operate on sparse graphs. While some GNN variants use edge features (Gilmer et al., 2017; Chen et al., 2019) or even compute dynamic edge states (Gong and Cheng, 2019), to the best of our knowledge no GNN architecture updates edge states based on the states of other edges as we do in Equations 5-8.

A number of studies of systematic generalization have examined the ability of neural networks to handle inputs consisting of familiar atoms that are combined in novel ways. Such research is typically conducted using carefully controlled train-test splits (Lake and Baroni, 2018; Hupkes et al., 2019; Sinha et al., 2019; Bahdanau et al., 2019; Keysers et al., 2020; Kim and Linzen, 2020). Various techniques have been used to improve generalization on some of the these benchmarks, including (but not limited to) data augmentation (Andreas, 2019; Akyürek et al., 2020), program synthesis (Nye et al., 2020), meta-learning (Lake, 2019), and task-specific model design (Guo et al., 2020). Most relevant to our work is the Neural Theorem Proving (NTP) approach proposed by (Rocktäschel and Riedel, 2017) and developed by (Minervini et al., 2020). NTP can be described as a differentiable version of Prolog backward-chaining inference with vector embeddings for literals and predicate names. Unlike NTP, Edge Transformers do not make rules explicit; we merely provide the networks

with an architecture that facilitates the learning of rule-like computations by the attention mechanism. Edge Transformers remain closer to general purpose neural models such as Transformers in spirit and, as demonstrated by our CFQ and COGS experiments, can be successfully applied in the contexts where the input is not originally formulated as a knowledge graph.

Finally, the Inside-Outside autoencoder proposed by Drozdov et al. (2019) for unsupervised constituency parsing is the most similar to our work at the technical level. The Inside-Outside approach also features vector states for every pair of input nodes. However, in contrast to the Edge Transformer, the attention is restricted by the span structure of the input.

## 6 Discussion

We present the Edge Transformer, an instantiation of the idea that a neural model may benefit from an extended 3D-tensor state that more naturally accommodates relational reasoning. Our experimental results are encouraging, with clear performance margins in favor of the Edge Transformer over competitive baselines on three compositional generalization benchmarks. Our ablations experiments confirm the importance of core intuitions underlying the model design.

Edge Transformers can be seen as belonging to the line of work on neuro-symbolic architectures (Mao et al., 2018; Rocktäschel and Riedel, 2017; Evans and Grefenstette, 2018, to name but a few). Compared to most work in the field, Edge Transformers are decidedly more neural: no logical gates or rule templates can be found in our architecture. We believe this is a promising direction for further exploration, as it allows the model to retain the key advantages of neural architectures: their generality, interchangeability and ability to learn from data.

Edge Transformer computations and memory usage scale as $O(n^3)$ with respect to the input size $n$. While this makes scaling to larger inputs challenging, in our experience modern GPU hardware allows training Edge Transformers with inputs of size up to a hundred entities. One way to use Edge Transformer in applications with larger inputs is to create Edge Transformer nodes only for the most salient or important components of an input. For example, in the language modelling context these could be just the heads of noun phrases and/or just the named entities. Future work could build and pretrain on raw text hybrid models that combine both Transformer- and Edge Transformer-based processing, the latter performed only for a small subset of nodes.

## Acknowledgments and Disclosure of Funding

We gratefully acknowledge the support of NVIDIA Corporation with the donation of two Titan V GPUs used for this research.

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
