# A   Additional Experiment Details

## A.1   CLUTTR and CFQ Dependency Parsing

The experiments were performed using a cluster of 12 GPUs (2 24GB, 2 12GB, 8 11GB). A single training run on CFQ or CLUTRR requires less than 30 minutes on a single GPU. Hyperparameters were selected using grid search. For both the Edge Transformer and (Relation-aware) Transformer models, hyperparameter search was the same. The number of units were varied between 200-400; batch size between 40 and 400 (for CLUTRR) and $5 \cdot 10^3$ and $1 \cdot 10^4$ words (for CFQ). Learning rates varied between $1 \cdot 10^{-4}$ and $2 \cdot 10^{-3}$, and the number of heads varied from between 4 and 8. For Transformer models, the number of layers varied from 5 to 8. For Edge Transformer models, the number of layers was varied from 6 to 8.

Hyperparameters for CLUTRR (for ET, baselines, and ablations) were optimized on the $k = 2, 3$ task, and fixed for the $k = 2, 3, 4$ task. For CFQ dependency parsing, hyperparameters were optimized on a $1\%$ random subset of the official random split (Keysers et al., 2020), and fixed for the MCD splits.

## A.2   CFQ Semantic Parsing

When tuning the Universal Transformer baseline we varied the number of layers from 2 to 6, the learning rate from 0.0001 to 0.001, the batch size of 64 to 128 and the number of epochs on the 20K random split training examples from 200 to 400. The number of heads was fixed to 8. For training on MCD splits with $\sim$90K training examples each we divided the number of epochs by 4 to keep the total number of training steps approximately the same. We found that longer training gave better results, hence all results in the paper are obtained with 100 training epochs.

## A.3   COGS Semantic Parsing

No hyperparameter search was performed for Edge Transformer on COGS. Architecture hyperparameters for Edge Transformer were matched to those of (Ontanón et al., 2021), who tuned the number of layers, hidden dimension, feed-forward dimension, and number of heads for their Transformer architectures. Their best Transformer model has two standard layers and a separate attention head for output, giving three QK attention layers total. We therefore use three layers for Edge Transformer. Default settings were used for optimizer hyperparameters.