# OpenReview forum: "Systematic Generalization with Edge Transformers"
_NeurIPS.cc/2021/Conference — NeurIPS 2021 Poster_

### Official Review · Reviewer_HbnY · 2021-07-10

**Rating:** 6
**Confidence:** 3

**Summary:**

Paper presents edge transformer, a novel transformer architecture that takes edge information, i.e. binary relation among tokens/entities as input. To further reduce the computational overhead, the authors observed that there is no need to attend on unrelated edges and therefore introduce triangular attention to replace vanilla attention. Experiments on two datasets (CLUTRR and CFQ) demonstrates the effectiveness of the proposed method.

**Limitations And Societal Impact:**

The authors addressed the limitations properly.

**Main Review:**

[Strengths]

[+] The paper is overall clear and well-written. The research problem here falls in the interest of the NeurIPS community and may have the potential impact on larger groups than NLP.

[+] The proposed improvement on transformers is well-motivated. I like the example in canonical deductive reasoning and I agree with this intuition. The method is technically sound and I cannot find any issues.

[+] The experiments can be limited but the superior results over other baselines are impressive.

[Weaknesses]

Having said those above, I do have some concerns given its current presentations and I feel this paper can be somehow premature for publishing:

[-] It is still unclear which part of this method contributes the most to the performances. The authors are suggested to conduct a comprehensive ablation study on their model. At least there are two factors to test on:

1. (major) vanilla transformer with relational input embedding (10)
2. (minor) triangular attention vs. vanilla self-attention

I'm particularly interested in the results of 1. Please try to include them in the rebuttal.

[-] I'm not that familiar with NLP datasets&benchmarks that focus on systematic generalization. But it seems that the two datasets in this paper do not even have a compositional generalization split and therefore the results could be less convincing given the authors' claim on improving compositionally. I will recommend including some results on the following datasets:

**SCAN** https://arxiv.org/abs/1711.00350

**COGS** https://arxiv.org/abs/2010.05465

**SyGNS** https://arxiv.org/abs/2106.01077

[-] There are more and more papers today that try to explain how transformers tackle the difficult downstream tasks by visualizing the attention matrix (ex. https://arxiv.org/abs/2104.14294). Given the illustration in sec. 2, I find it could be necessary to visualize the resulting attention over all the edges. The authors could try to answer this question in their inspection: Do the proposed transformers attend to the expected edges among input tokens?

[Suggestions]

I'll be willing to increase my score if the following points can be addressed in a rebuttal:

- ablation study on input embedding and the proposed attention mechanism.
- results of at least one aforementioned dataset.
- visualization on the attention matrix and some explanations.

**Time Spent Reviewing:**

2.5

---

> ### Author Response · Authors · 2021-08-10
> **Compositional generalization datasets; baselines; attention**
>
> We thank the reviewer for their work and for the suggestions they have made.
>
> **But it seems that the two datasets in this paper do not even have a compositional generalization split**
>
> We will improve the description of the datasets in the revised version. All results that we report in the paper are obtained on compositional splits of the CLUTRR and the CFQ benchmarks respectively. These two datasets were introduced specifically to probe the ability of models to generalize compositionally. CLUTRR achieves this by testing on reasoning chains that are longer than in training (lines 148 - 156) and CFQ splits the data into train and test by maximizing compound divergence (lines 173 - 179). Roughly, test and train have high compound divergence if test contains the same distribution over individual words as train, but combines them in novel ways. While SCAN, COGS, and SyGNS are also interesting datasets for compositional generalization, their output representations do not have a direct interpretation as a set of edges and thus they are not directly modelable in the current Edge Transformer framework.
>
> We expect to have access to a dependency parsing version of COGS soon, and will evaluate Edge Transformer on this dataset once it is available.
>
> **The authors are suggested to conduct a comprehensive ablation study on their model. At least there are two factors to test on:
> (major) vanilla transformer with relational input embedding (10)**
>
> We will improve our description of the Relation-aware Transformer, which is used as a baseline for CLUTRR. This model uses a relational input embedding, and combines these embeddings with a vanilla transformer. We are happy to implement other variants of this model if useful.
>
> **(minor) triangular attention vs. vanilla self-attention**
>
> Since submitting the paper, we have implemented a new baseline which uses vanilla self-attention on edges. Please see the description of Flattened Edge Transformer in the General Response to Reviewers. This baseline performs worse than Edge Transformer on both CLUTRR and CFQ.
>
> **There are more and more papers today that try to explain how transformers tackle the difficult downstream tasks by visualizing the attention matrix.**
>
> We thank the reviewer for the helpful suggestion regarding visualization. The attention mechanism for an Edge Transformer is a 3d tensor for each head, making visualization difficult. While making revisions, we will consider whether there are interesting 2d projections that illustrate the way that the model computes transitive closures.

---

> > ### Comment · Reviewer_HbnY · 2021-08-21
> > **Response to the rebuttal.**
> >
> > I would like to thank the authors for their detailed clarification. It seems that the presentation did confuse me a lot. I hope the author can revise their paper to reflect those aforementioned changes been made. I've increased my score to 6.

---

### Official Review · Reviewer_ozBq · 2021-07-12

**Rating:** 7
**Confidence:** 4

**Summary:**

This paper introduces a variation of the Transformer architecture where attention happens between edges as opposed to nodes. As such, the input and output to the Edge Transformer are 3D $d \times n \times n$ tensors instead of 2D $d \times n$ tensors. This new architecture, together with its novel “triangular attention” mechanism, are introduced with the objective of favoring generalization in compositional reasoning. Two sets of experiments validate the architecture’s benefits over previous architectures.

**Limitations And Societal Impact:**

Limitations have been addressed.

**Main Review:**

The architecture and the proposed triangular attention appear to be a relatively natural variation of Transformers, but they are novel. They are well motivated by the problem of improving compositional generalization. The paper is very clear and has all the details needed to replicate the architecture. Overall, it is a welcome addition to the literature.

Some details are lacking in the experiments section, especially with respect to the dependency parsing task (the main reference  being an unspecified paper): Is the task semantic parsing or some variation of it? What are examples of input and output? Why is this task supposed to measure compositional generalization? Expanding on these points would help.

In general, the set of experiments is fairly limited and does not appear to provide enough evidence that the Edge Transformer could be more suitable than the ordinary Transformer for solving more complex tasks (especially due to the significant computational increase). The computational disadvantage of the Edge Transformer is acknowledged in Section 6, where the authors also provide ideas on how this issue could be addressed in tasks like language modeling. Additional experiments would be very welcome, but they do not necessarily fall within the scope of this paper.


**Time Spent Reviewing:**

2

---

> ### Author Response · Authors · 2021-08-10
> **Dependency parsing clarifications**
>
> We thank the reviewer for their encouraging comments and suggestions.
>
> **Some details are lacking in the experiments section, especially with respect to the dependency parsing task (the main reference being an unspecified paper)**
>
> Thank you for these questions. We will clarify the details of the task.
>
> **Is the task semantic parsing or some variation of it? What are examples of input and output?**
>
> It is a syntactic dependency parsing dataset which was generated from the original CFQ semantic parsing dataset. Labels are provided for every pair of words. For a pair (x,y), the label indicates whether word y is a dependent of word x (or vice-versa), and if so, what type of dependency holds between the words. The label set is taken from Universal Dependencies, and includes relations such as subject, object, and modifier. Raw sentences are provided as input, and the task is to predict each pairwise label.
>
> **Why is this task supposed to measure compositional generalization?**
>
> The CFQ dataset contains test/train splits that maximize “compound divergence“ -- a measure introduced to quantify compositional generalization. Roughly, test and train have high compound divergence if test contains the same distribution over individual words as train, but combines them in novel ways. The dependency parsing version of CFQ that we use contains the same questions that are split into train and test in the same way. The only difference is the annotation used: syntactic dependency tree instead of the SPARQL query. The compositional generalization aspect of the task is fully retained.  We use the dependency version of the CFQ tasks to showcase Edge Transformer’s ability to generalize compositionality when predicting graph-structured outputs.

---

### Official Review · Reviewer_V441 · 2021-07-17

**Rating:** 6
**Confidence:** 3

**Summary:**

The authors propose Edge Transformers to solve  CLUTRR relational reasoning challenge and Compositional Freebase Questions benchmark. The edge transformer is based on triangular attention which is weighted sum of the representations of all the triangular nodes. The proposed method can achieve better performance than the baselines of Transformer, LSTM, and GAT.

**Limitations And Societal Impact:**

A little bit overclaim the novelty of edge-transformer, but overlook related works on graph neural networks.

**Main Review:**

Overall, the proposed method is interesting. I didn't know other works focusing on weighted sum of the representations in triangular. However, this method is very close to combining GNN and Transformer. How about first use GAT to encode the graph and then run another transformer layer on all the nodes? I would like to see the performance of this baseline. I feel the authors overplay the model structure but ignore the simplest way to combine GAN and Transformer.

Other comments:
1. For dependency parsing tasks, it would be better to test the method on more benchmark datasets. Or reimplement some benchmark baselines with graph neural networks. Vanilla LSTM and Transformer are too weak baselines.
2. The intuitions section doesn't seem novel enough. The motivation is close to graph neural networks, which can also merge the information of the nodes with relation.s

**Time Spent Reviewing:**

1

---

> ### Author Response · Authors · 2021-08-10
> **Baselines; CFQ dataset**
>
> We thank the reviewer for their encouragement and suggestions.
>
> **How about first use GAT to encode the graph and then run another transformer layer on all the nodes? I would like to see the performance of this baseline.**
>
> We will improve the description of the Relation-aware Transformer baseline in the revised version. This model embeds edges from the input graph, and combines these embeddings with a Transformer. This is similar to the baseline being proposed, but we would be happy to run other baselines if it is useful.
>
> **For dependency parsing tasks, it would be better to test the method on more benchmark datasets.**
>
> We chose CFQ for the dependency parsing task, because it has a train/test split which evaluates compositional generalization. At the moment, we have no other benchmark for evaluating this in dependency parsing. COGS is an alternative dataset which was designed for evaluating compositional generalization, but it is currently formulated as a seq2seq task. However, we expect to receive a dependency parsing version of this dataset soon, and will test Edge Transformer on it in our future work.
>
> **Or reimplement some benchmark baselines with graph neural networks. Vanilla LSTM and Transformer are too weak baselines.**
>
> It is not straightforward to apply standard GNNs to CFQ dependency parsing, since the input for this task is not a graph, but a sequence of words. However, since submission, we have evaluated both CLUTRR and CFQ on a new GNN baseline which can be adapted for the task. Please see the general response to reviewers for a complete description of this GNN (Flattened Edge Transformer).
>
> This GNN baseline performs worse than Edge Transformer on both CLUTRR and CFQ.

---

> ### Author Response · Authors · 2021-09-01
> **New results**
>
> Dear Reviewer V441,
>
> We hope that you are doing well. In your review, you wrote:
>
> > For dependency parsing tasks, it would be better to test the method on more benchmark datasets
>
> Since submission, we have performed experiments on a new task, CFQ semantic parsing, described in more detail [here](https://openreview.net/forum?id=UUds0Jr_XWk&noteId=FPDeMlFr1W) and [here](
> https://openreview.net/forum?id=UUds0Jr_XWk&noteId=QDA2iaBcEma). For this task, we were able to compare Edge Transformer to baselines from previously published work.
>
> Does this address your concern about testing Edge Transformer on a new dataset and against stronger baselines?

---

### Official Review · Reviewer_Ecbf · 2021-07-17

**Rating:** 6
**Confidence:** 4

**Summary:**

The paper presents a transformer-like model that carries vectors on graph edges instead of the nodes, and updates the vectors through a new (triangular) formulation of self-attention. The Edge Transformer demonstrates better generalization than baselines on two relational tasks.

**Limitations And Societal Impact:**

Yes

**Main Review:**

- Originality

Section 5 provides a useful summary of prior work involving edge vectors in transformers and GNNs. The proposed form of triangular attention appears to be both novel and well-motivated.


- Quality

The Edge Transformer (ET) is evaluated against two baseline models, Relation-aware Transformer (RT) and a basic Transformer, on two supervised tasks (CLUTRR and CFQ) as summarized in Table 1. The results in Figure 3 and Table 2 indicate that ET outperforms the baselines on these tasks.

Section 4.1 adequately describes the CLUTRR (relational reasoning) task, and the process of applying ET to it, by embedding the edge labels to obtain the initial edge vectors, and retrieving the predicted class from the query edge output vector. But the corresponding details are missing for the RT baseline model. RT carries vectors on both nodes and edges, but updates only the node vectors. Line 162 notes that RT “can be conditioned on an arbitrary graph”, so perhaps its edge vectors are initialized in the same way as those of ET, and presumably RT’s node vectors are initialized to zero. But then what happens? Since RT never updates its edge vectors, and the query edge vector is initialized to some non-informative value, retrieving the predicted class from the unchanged query edge vector could only produce a fixed output. But Figure 3 clearly shows that RT performs better than chance. It appears that RT was somehow adapted to this task, but the paper does not explain how. Without understanding how the baseline model was actually used, we cannot evaluate the significance of the difference in performance between ET and RT on the CLUTRR task. In addition, since RT was not designed to update its edge vectors, a more appropriate baseline model for this task would probably be some GNN that was designed to carry edge vectors and update them in each round of message passing.

Section 4.2 describes the CFQ (dependency parsing) task, where the input is a sequence of word tokens, and the output is a label for each token pair. As with the CLUTRR task, no detailed description is provided for how the pairwise output labels are obtained from the baseline transformer, which carries no edge vectors. The results in Table 2 indicate that ET’s edge-based triangular attention delivers improved performance over standard transformer self-attention in this task.

The paper’s contribution would be strengthened by positive results on a third task. In particular, it seems important to know how well ET performs as a simple language model. It should be relatively straightforward to train ET on Wikitext and generate a training curve showing perplexity improvement over epochs. Alternatively, ET could be evaluated against GNNs on a task that requires reasoning over edge features.


- Clarity

As noted above, the paper is missing essential details about how the baseline models were applied to the tasks. Apart from this shortcoming, the paper is quite clear and well-written. Section 3 in particular is excellent. In section 4.3, the term ‘ablation’ would be more readily understood than ‘lesion’.


- Significance

Given the successes of transformers in NLP and beyond, any fundamental improvement to the transformer architecture can lead to major benefits.


- Conclusion

I view this as a potentially important work that currently lacks sufficient experimental support. I do not believe that novel advances such as triangular attention should be required to achieve SOTA results on major problems prior to publication, but I do believe that carefully described baselines are essential to any meaningful evaluation.

- POST-DISCUSSION UPDATE

To their credit, the authors have remained deeply engaged throughout the review process, and have greatly improved the work. I still have reservations about the CLUTRR task baseline, but this seems less important than the positive results on the newly added semantic parsing task.

If the original experimental section had not left out so many key details, I would have more confidence that the next, significantly revised version of the experimental section would cover all that is required. But I’m raising my rating by 2 points, and petitioning the authors to seek feedback from someone unfamiliar with the work to help bring the experimental section up to the same standard as Section 3.


**Time Spent Reviewing:**

8

---

> ### Author Response · Authors · 2021-08-10
> **Model details; baselines; preliminary seq2seq results**
>
> Thank you for your detailed and constructive review. We agree that the paper would benefit from including more details on the baselines and we will add this information in the next revision. For now, please find your questions answered below:
>
> **Since RT never updates its edge vectors, and the query edge vector is initialized to some non-informative value, retrieving the predicted class from the unchanged query edge vector could only produce a fixed output. But Figure 3 clearly shows that RT performs better than chance. It appears that RT was somehow adapted to this task, but the paper does not explain how.**
>
> Predictions for the query are computed by concatenating the vectors for the two nodes that occur in the query, and running this through a linear decoder.
>
> **In addition, since RT was not designed to update its edge vectors, a more appropriate baseline model for this task would probably be some GNN that was designed to carry edge vectors and update them in each round of message passing.**
>
> Since submission, we have carried out experiments on two new baselines with updating edge vectors. Please see our General Response to Reviewers for more details. We are happy to consider additional models if useful.
>
> **Section 4.2 describes the CFQ (dependency parsing) task, where the input is a sequence of word tokens, and the output is a label for each token pair. As with the CLUTRR task, no detailed description is provided for how the pairwise output labels are obtained from the baseline transformer, which carries no edge vectors.**
>
> For the baseline transformer, pairwise output labels are obtained by passing each pair of nodes through a biaffine layer, following Dozat & Manning (2017). Separate biaffine weights are used to predict labeled and unlabeled attachment. This is the same output layer that was used for the BiLSTM baseline. See: https://github.com/stanfordnlp/stanza/blob/main/stanza/models/depparse/model.py#L136
>
> **In particular, it seems important to know how well ET performs as a simple language model.**
>
> Edge Transformer in its current formulation is not well suited for long-range language modelling because of the O(n^3) computational complexity. That being said, in our ongoing investigations we are experimenting with a seq2seq version of the EdgeTransformer that adds causal (autoregressive) masking to the triangular attention mechanism. For these experiments we use the original version of the CFQ benchmark where the predicted output is a SPARQL query that is represented as a sequence of tokens. In this setting, we observe performance and learning curves similar to those of the vanilla Transformer model. This suggests that EdgeTransformer inherits the universality characteristic of the Transformer, but can generalize better when provided appropriate training signal, as illustrated by our CLUTRR and CFQ dependency parsing experiments.

---

> > ### Comment · Reviewer_Ecbf · 2021-08-20
> > **Question**
> >
> > Thank you for the explanation of how predictions are obtained from RT on the CLUTTR task. How are its node and edge vectors initialized?

---

> > > ### Author Response · Authors · 2021-08-20
> > > **RT node and edge initialization**
> > >
> > > We experimented with zero and random initialization of RT node vectors. Performance was better with random initialization, so this is what is reported in the paper.
> > >
> > > Edge vectors are initialized with learned embeddings. For edge (i,j), either nodes i and j have an observed relation, or no relation is observed. Separate embeddings are learned for each relation and the no-relation case. The same edge initialization is used for Edge Transformer.

---

> > > > ### Comment · Reviewer_Ecbf · 2021-08-20
> > > > **Question**
> > > >
> > > > So let me try to summarize your application of the RT baseline. CLUTRR is a graph-structured edge-prediction task with input features on the edges but not the nodes. As a transformer, RT consumes node features and produce node vectors as outputs. RT differs from a normal transformer in that it can also accept edge features, but it does not update the edge vectors. So on the CLUTRR task, RT must somehow learn to transfer information from its static edge vectors to its node vectors, then perform the usual multi-layer computation over those nodes, and produce output node vectors which are then concatenated by pairs and projected to produce the prediction for each edge. Is this an accurate high-level description?

---

> > > > > ### Author Response · Authors · 2021-08-20
> > > > > **RT summary**
> > > > >
> > > > > Yes that is correct.
> > > > >
> > > > > As described in the general response to reviewers, we have also implemented a variant of RT, in which edge vectors are also updated in each layer. In this case, the prediction for each edge is computed directly from the edge vector (rather than concatenating node vector pairs).

---

> > > > > > ### Comment · Reviewer_Ecbf · 2021-08-20
> > > > > > **RT on CLUTRR**
> > > > > >
> > > > > > Yes, I think your variant of RT that updates its edge vectors would be a much more meaningful baseline model for this task. Another excellent baseline model would be a non-transformer GNN/MPNN that updates its edge vectors (Neural Message Passing for Quantum Chemistry, Gilmer et al. 2017).

---

> > > > > > > ### Author Response · Authors · 2021-08-20
> > > > > > > **Edge updating models**
> > > > > > >
> > > > > > > **Edge-updating RT results**
> > > > > > > We will include the RT variant that updates edge vectors in our revised paper. After hyperparameter optimization, we obtained the following results for this baseline:
> > > > > > >
> > > > > > > CLUTRR accuracy (Train k=2,3). Edge Transformer: 0.74; edge-updating RT: 0.63
> > > > > > >
> > > > > > > CLUTRR accuracy (Train k=2,3,4). Edge Transformer: 0.90; edge-updating RT: 0.83
> > > > > > >
> > > > > > > **Non-transformer baseline** Thank you for the suggestion to use a non-transformer GNN with updating edge vectors. We have read through the Gilmer et al. paper, and it appears that the primary model does not update its edge vectors. Would you be able to clarify which model to use?
> > > > > > >
> > > > > > > **Additional GNN baseline** In addition to this baseline, we will also implement a sparse GNN version of the edge-updating RT. Nodes will only communicate with nodes that share an observed edge.

---

> > > > > > > > ### Comment · Reviewer_Ecbf · 2021-08-20
> > > > > > > > **MPNNs**
> > > > > > > >
> > > > > > > > True enough, but Gilmer et al. is the most frequently cited work for MPNNs in general, and the paper explains that "one could also learn edge features in an MPNN by introducing hidden states for all edges in the graph and updating them analogously to equations 1 and 2. Of the existing MPNNs, only Kearnes et al. (2016) has used this idea."

---

> > > > > > > > > ### Author Response · Authors · 2021-08-21
> > > > > > > > > **MPNNs**
> > > > > > > > >
> > > > > > > > > Thank you for pointing us to this description. We will implement a variant of Gilmer et al.'s model in which edges are updated as suggested. We expect to have results within the next 24 hours.

---

> > ### Comment · Reviewer_Ecbf · 2021-08-21
> > **After studying the authors' responses**
> >
> > The authors have adequately addressed my questions about how their baseline models were used. These details improve the quality of the paper, but I still share the concerns expressed by other reviewers that experimental support is lacking.
> >
> > Only two tasks are considered. On the CLUTRR task, the RT model seems to be more of a sanity check than a meaningful baseline. The RRT model introduced by the authors in their response is a more credible baseline than RT, but it is still a novel, unvetted model. The most appropriate baseline for this graph-structured task would be a GNN from the literature that updates its edge vectors. Without such a baseline for comparison, the CLUTRR task doesn’t provide much evidence to help us evaluate the significance of the Edge Transformer’s contribution. The only other experiment performed is dependency parsing (CFQ), for which the anonymous baseline transformer is hard to assess.
> >
> > For these reasons, my original rating remains unchanged. But whether this paper is accepted or not, I look forward to the exciting line of work described by the authors involving a seq2seq version of the Edge Transformer, where the O(n^3) cost will not prevent its application to more general language modeling.

---

> > > ### Author Response · Authors · 2021-08-30
> > > **GNN baseline**
> > >
> > > The reviewer requests that we implement a GNN with updating edges, stating:
> > >
> > > > The most appropriate baseline for this graph-structured task would be a GNN from the literature that updates its edge vectors. Without such a baseline for comparison, the CLUTRR task doesn’t provide much evidence to help us evaluate the significance of the Edge Transformer’s contribution.
> > >
> > > At the [reviewer's suggestion](https://openreview.net/forum?id=UUds0Jr_XWk&noteId=S-WK0HzxNQB), we selected the highest-performing model from Gilmer et al. (2017), and ran it on the edges of the CLUTRR graph. The model operates on the [line-graph](https://pytorch-geometric.readthedocs.io/en/latest/modules/transforms.html#torch_geometric.transforms.LineGraph) of the CLUTRR graph. Edges from the original graph are turned into nodes, and are adjacent when they share a node in common in the original graph.
> > >
> > > This GNN performs worse than Edge Transformer on CLUTRR. After hyperparameter search, the best performing model achieves accuracy of 0.55 on the k=2,3 task (vs. 0.74 for Edge Transformer), and 0.67 on k=2,3,4 (vs. 0.90 for Edge Transformer).

---

> > > > ### Comment · Reviewer_Ecbf · 2021-08-30
> > > > **GNN baseline**
> > > >
> > > > I admire your hard work, but my suggestion was to use a GNN from the literature that updates its edge vectors, such as the one by Kearnes et al. where edge states encoded graph distances between atoms/nodes. Your new baseline model here is very different; it first converts edges into nodes, then operates on the new graph. Since this is a novel, unvetted model, I don’t think it provides a credible baseline for your CLUTRR experiments. Many recently published, competitive GNNs that perform edge property prediction can be found on this leaderboard:  https://ogb.stanford.edu/docs/leader_linkprop/

---

> > > > > ### Author Response · Authors · 2021-08-31
> > > > > **Request for clarification**
> > > > >
> > > > > Thank you for your response.
> > > > >
> > > > > CLUTRR contains no node information in the input or labels, only edge information. We therefore could not have used the Kearnes et al. model without modification; the Kearnes model takes as input both atom features (node-level) and atom pair (edge-level) features, and combines them.
> > > > >
> > > > > Could you suggest a specific model that we can use without modification?

---

> > > > > > ### Comment · Reviewer_Ecbf · 2021-08-31
> > > > > > **GNN baseline**
> > > > > >
> > > > > > Absence of node features would not require modification of the Kearns model; you can use zero vectors as inputs to the nodes. But the Kearns model is several years old. It will take some investigation to find a competitive GNN that is used today on edge property prediction tasks. The Open Graph Benchmark leaderboard is a good place to begin the search.

---

> > > > > > > ### Author Response · Authors · 2021-08-31
> > > > > > > **Thank you for clarification**
> > > > > > >
> > > > > > > Thank you for your clarification. We will implement the model by Kearnes et al 2016 as described in Section 2 of (Gilmer et al, 2017). We will initialize the node states with zero vectors as you suggested.
> > > > > > >
> > > > > > > We have also inspected a number of papers that Stanford leaderboard refers to and did not find any GNNs with edge updates in them. We would appreciate if you can point us to a specific one.
> > > > > > >
> > > > > > > Meanwhile, we would like to hear your opinion on the state-of-the-art CFQ semantic parsing results that we reported in a general message to all reviewers yesterday.

---

> > > > > > > > ### Comment · Reviewer_Ecbf · 2021-08-31
> > > > > > > > **GNN baseline**
> > > > > > > >
> > > > > > > > It's true that GNNs are evolving rapidly. Since CLUTRR is an edge property prediction task, it can be approached through updates to edge vectors. But whether the GNNs on today's OGB leaderboard use that approach or not, they are still the most appropriate GNNs to use as baselines on a task like CLUTRR.

---

### Author Response · Authors · 2021-08-10
**General response to reviewers**

We thank the reviewers for their helpful comments. Several reviewers have suggested that the results would be strengthened with additional baselines. We have therefore implemented two additional baseline models, and compared the results with Edge Transformer.

1. We have implemented a modified version of Relation-aware Transformer (RT), which we call Relation-updating Relation-aware Transformer (RRT). In each layer, both node vectors and edge vectors are updated (in contrast to RT, in which only nodes are updated). The node updates remain the same as in standard RT. Each edge (x,y) is updated by concatenating node representations for x and y, and passing this through a linear layer. Edge vectors are passed through residual, normalization, and feedforward layers which are identical to standard RT.

2. We have implemented a “flattened” version of Edge Transformer, which can be viewed as a GNN on edges. Adjacencies between edges are the same as Edge Transformer: edge (x,y) is adjacent to edges (x,z) and (z,y), for all z. Instead of triangular attention, edges are updated using vanilla attention. All other details of the architecture remain the same.

Both baselines perform worse than Edge Transformer on CLUTRR, and Flattened Edge Transformer performs worse on CFQ (RRT cannot be straightforwardly applied to CFQ, as the input for CFQ is not a graph). The same hyperparameter search was performed for these architectures as for the other models.

On CLUTRR (Train k=2,3) average test accuracy is:

Edge Transformer: 0.74;
RRT: 0.63;
Flattened Edge Transformer: 0.49

On CLUTRR (Train k=2,3,4) average test accuracy is:

Edge Transformer: 0.90;
RRT: 0.83;
Flattened Edge Transformer: 0.56.

On CFQ (MCD-3) test LAS is:

Edge Transformer: 95.2;
Flattened Edge Transformer: 91.1

---

### Author Response · Authors · 2021-08-19
**Further reviewer questions**

Dear reviewers, We hope that you are doing well. We wanted to check whether there are any further questions that we can answer for you. Thank you.

--Paper 9915 Authors

---

### Author Response · Authors · 2021-08-29
**CFQ Semantic Parsing Results**

Dear reviewers,

Several of you suggested that the paper could be strengthened by adding results on another task. We would like to share with you the results of new experiments that we have conducted on the original seq2seq semantic parsing version of the CFQ compositional generalization task.

**TL;DR:** Edge Transformer achieves the SoTA performance among non-pretrained generic models on the CFQ semantic parsing task.

**Longer version:**

The table below summarizes the average performance on MCD splits that is further averaged over 5 (for Edge Transformer) or 30 (for Universal Transformer) seeds:

Model       &emsp;     &emsp; &emsp; &emsp; &emsp; &nbsp; &nbsp;  &nbsp; &nbsp; &nbsp; &nbsp; &nbsp; &nbsp; &nbsp; &nbsp; &nbsp;           | mean +- std (%)

___

Universal Transformer (ours)  &emsp;  | 21.26 +- 1.06

**Edge Transformer      &emsp;  &emsp; &emsp;   &emsp;    &nbsp;&nbsp;&nbsp;  	 | 24.69 +- 1.27**

___

Universal Transformer* 	&emsp; &emsp;  &nbsp; &nbsp;&nbsp;  | 18.9 +- 1.13

 (Keysers et al, 2020)	&emsp;   &emsp; &emsp;  &nbsp; &nbsp; &nbsp; &nbsp;&nbsp;        	|

Evolved Transformer* 	&emsp; &emsp;  &emsp; &nbsp;&nbsp; | 20.8 +- 0.56

 (Furrer et al, 2020) 	&emsp;   &emsp;  &emsp; &emsp; &nbsp;&nbsp; &nbsp;       	|

T5-small w/o pretraining * &emsp; &emsp;     | 21.4 +- 1.2

 (Furrer et al, 2020) 	&emsp; &emsp;  &emsp; &emsp; &nbsp;&nbsp; &nbsp;          	|

-----------------------------------------------------------

\* Furrer et al and Keysers et al report 95% confidence interval obtained from n=5 runs. We convert these results to mean +- std by using the following Python expression:
std_deviation = interval_width * n ** 0.5 / -scipy.stats.t.ppf(0.025, n – 1)

According to the t-test and permutation test, the Edge Transformer results are significantly better than Universal Transformer ones (both tests give p-values <= 0.00001). To our knowledge this is the first time an improvement on CFQ is achieved by changing the core model architecture and without adding any highly CFQ-specific inductive bias.

In addition to improvements in compositional generalization, we have observed a significant boost in sample efficiency. On a reduced data version of the CFQ random split with 20K instead of 100K training examples, Edge Transformer learns faster and achieves 99.08 +- 0.1 accuracy, whereas the best Universal Transformer hyperparameter setting peaked  at 98.45 +- 0.08.

The encoder-decoder version of Edge Transformer that we used in these experiments is very similar to what we describe in the paper. The main difference is the addition of causal masking to the triangular attention mechanism. We also found it necessary to use d_ff = 4d hidden units in the hidden layer of the fully-connected network (FFN in Eq. 3) and to warm up the learning rate. Both techniques are routinely used for Transformer models, and apart from adopting them, we obtained the above results with almost no hyperparameter tuning. The experiments however took a significant amount of time and compute, as we had to train the cubic-time EdgeTransformer model on longer sequences (which is why we first performed experiments on the dependency parsing version of the CFQ challenge).

We believe these results strengthen the paper’s claim: Edge Transformer exhibits significantly higher compositional generalization than Transformer. As suggested by Reviewer Ecbf, any fundamental improvement to the Transformer architecture can lead to major benefits. We thus believe this paper will be of great interest to NeurIPS attendees, as it proposes and experimentally validates a promising direction for improving Transformers. While the O(n^3) computational complexity is a practical challenge to be addressed, we think it would be reasonable to leave it to future work.

---

> ### Comment · Reviewer_Ecbf · 2021-08-30
> **New task?**
>
> Can you describe the relationship between this additional "seq2seq semantic parsing version of the CFQ compositional generalization task" and the CFQ task described in your paper? Are they both edge-prediction tasks? Or are they in fact the very same task?

---

> > ### Author Response · Authors · 2021-08-30
> > **CFQ semantic parsing vs. dependency parsing**
> >
> > The new results are on the CFQ semantic parsing task, which is different than the dependency parsing task already presented in the paper. Semantic parsing is the original task from Keysers et al. (2020). The dependency parsing task was generated from the Keysers et al dataset, by deterministically converting semantic parses into dependency parses.
> >
> > For CFQ semantic parsing, the input is an English question, and the goal is to predict the semantic parse of this question, which is a SPARQL query. For example,
> >
> > **Question**:  Did Agustin Almodovar executive produce Deadfall
> >
> > **SPARQL query**: SELECT count(*) WHERE {\nns:m.04lhs01 ns:film.producer.films_executive_produced ns:m.0gx0plf\n}
> >
> > The query is tokenized, and the task is to predict the query from the question, as in a standard seq2seq setup. We used causal masking for Edge Transformer to allow it to perform seq2seq. Unlike the dependency parsing task, no edge supervision is provided.
> >
> > In our [earlier response](https://openreview.net/forum?id=UUds0Jr_XWk&noteId=FPDeMlFr1W), Edge Transformer results are compared to previously published results on CFQ semantic parsing, from Keysers et al. (2020) and Furrer et al. (2020).

---

> > > ### Comment · Reviewer_Ecbf · 2021-08-31
> > > **New task**
> > >
> > > Your results on this new task (CFQ semantic parsing) seem to strengthen the case for the usefulness of the Edge Transformer. But an accurate review of this new work would require a more detailed explanation in a new experimental section of the paper, which could be provided through an anonymous link to a pdf. I don’t know whether the other reviewers would want to consider such material at this late stage of the discussion period. And I’m not sure whether such an expansion of your submission is appropriate. I have raised this question with the AC and the other reviewers.

---

> > > > ### Author Response · Authors · 2021-08-31
> > > > **On CFQ semantic parsing experimental details and on CFQ dependency parsing baselines**
> > > >
> > > > Thank you for continued engagement in this discussion.
> > > >
> > > > We have sent a general message to all reviewers with additional experimental details on our Edge Transformer semantic parsing experiments. Please let us know if have you any questions.
> > > >
> > > > We would also like to revisit your concerns regarding our dependency parsing experiments. To limit the number of conversation threads, we will ask our question here. In your recent message you say:
> > > >
> > > > > The only other experiment performed is dependency parsing (CFQ), for which the anonymous baseline transformer is hard to assess.
> > > >
> > > > Can you please clarify what you mean by “anonymous baseline transformer”? We are not using pretrained transformers such as BERT, T5 or GPT, because that would be an unfair baseline to our Edge Transformer model that we train from scratch.

---

> > > > > ### Comment · Reviewer_Ecbf · 2021-09-01
> > > > > **CFQ**
> > > > >
> > > > > Good point. I should have said "The only other experiment performed is dependency parsing (CFQ), a task proposed in unpublished work, and using as baselines a BiLSTM model from that work and a transformer. This reliance on anonymous work makes the Edge Transformer's performance on the CFQ task hard to assess."

---

### Author Response · Authors · 2021-08-31
**Additional experimental details for CFQ semantic parsing experiments**

Upon the request of Reviewer Ecbf, we are sharing here additional experimental details for the CFQ semantic parsing experiments.

# Approach
In adapting the Edge Transformer to generate a sequence of tokens we follow the original Transformer paper by Vaswani et al, 2018. First, an encoder Edge Transformer processes the question and produces a 3D tensor state $x_{enc}$. Next, the decoder Edge Transformer generates output tokens one by one, left-to-right. To produce the output distribution for the token $i$ we feed the state of the loop edge $(i, i)$ through a linear layer. To connect the decoder to the encoder we insert $x_{enc}$ into the decoder’s initial state. This operation is the Edge Transformer’s equivalent of how the Transformer decoder cross-attends to the Transformer encoder’s hidden states. To compute training log-likelihoods we adapt the Transformer decoder’s causal masking approach to our triangular attention mechanism (<10 lines of code).
# Implementation and Training details
We implement semantic parsing on CFQ with the same Huggingface-based codebase for both Universal Transformer and the Edge Transformer. The sequence generation code is the same for both models.

As suggested in the original CFQ paper by Keysers et al, we tune the hyperparameters on the random split and keep the compositional generalization splits for testing. We reduce the size of the random split training data from 100K examples to 20K to increase the sensitivity to different hyperparameter choices. We tuned the Transformer baseline along the following axis
- weight-sharing across layers (i.e. Transformer vs Universal Transformer)
- number of layers (2, 4, 6)
- learning rate (0.0001 -  0.001)
- batch size (64, 128)
- number of epochs (200, 400)

We found that 6 layer Universal Transformers trained with the learning rate of 0.0006 for 400 epochs perform the best (98.45% accuracy).

We performed very little hyperparameter tuning for Edge Transformer because the Edge Transformer experiments were very computationally intensive (which is why we first did the dependency parsing experiments). Each training run took 15-30 hours of training on 4 GPUs with 32gb each. We found that using the same hyperparameter settings as the ones we used for Universal Transformer (except for using 4 layers instead of 6 to save time) allows us to reach a higher random split accuracy: 99.08%.

Finally, we trained Universal Transformer (30 times) and Edge Transformer (5 times) on each of the three MCD splits. Notably, our Universal Transformer baseline matches the best reported results for models that are trained from scratch (Furrer et al, 2020) with the average accuracy of 21.26+-1.06%. The Edge Transformer reaches a higher average accuracy of 24.69+-1.27%.

We would be happy to answer any questions.

---

> ### Comment · Reviewer_Ecbf · 2021-09-01
> **Details of new task**
>
> This is a clearer description, thank you. Good job on tuning the baseline's hyperparameters. Do you plan to make all of this code available?

---

> > ### Author Response · Authors · 2021-09-01
> > **Code availability**
> >
> > We are in the process of anonymizing our code. We will release it just as we have already released the code for our other experiments (see supplementary material).

---

### Decision · Program_Chairs · 2021-09-27

**Decision:**

Accept (Poster)

**Comment:**

This paper proposes 'Edge Transformer' which is a Transformer architecture that assigns an embedding to the edge between two tokens. A 'triangular attention' is used to update the edge embeddings, which basically computes, for edge a-b, a weighted combination of the product of edge embeddings a-n * n-b for each intermediary node n (N^3). The authors test this method on systematic generalization datasets such as CLUTTR (link prediction) and on a subtask derived from CFQ (dependency label prediction).

--

The paper was generally found clear, although some concerns about the experimental setup need to be addressed in the revision. A reviewer found improvements over baselines for systematic generalization to be "impressive". There has been an extensive back-and-forth between reviewers and authors on the validity of the baselines for the task considered. I thank both reviewers and authors for their fruitful exchange. The authors provided new positive results for the baselines suggested by reviewers. Initial concerns rotated around the synthetic nature of the task considered: the authors improved on this point by providing new results on the full semantic parsing task of CFQ by testing on each MCD split: the Edge Transformers outperforms carefully tuned previous (non pre-trained) transformers for this task. These results must still be taken with a grain of salt, given that pre-trained models achieve better results. However, these results strengthen the paper considerably. One suggestion I can think of in this respect is whether the authors could try to start from a pre-trained T5 model, add parameters for their edge attention, and fine-tune everything on the downstream CFQ task. This could maybe give a better idea on whether this model holds promise in the general setting, e.g. gains can be obtained without retraining the all model from scratch. If the potential impact of novel attention mechanism can extend to a large set of tasks, still major concerns rotate around the computational complexity of this approach (N^3) which makes it hardly applicable in general.

Overall, the reader might be left to wonder whether this method is applicable or is beneficial when it comes to large-scale pre-training of Transformer architectures. Nevertheless, provided that the authors integrate all the precious reviewers' feedback and the new experimental results in the final version of the paper, reviewers and myself agree that this paper can make an interesting addition to the conference.